# Disconnected Manifold Learning for Generative Adversarial Networks

**Mahyar Khayatkhoei**
Department of Computer Science
Rutgers University
m.khayatkhoei@cs.rutgers.edu

**Ahmed Elgammal**
Department of Computer Science
Rutgers University
elgammal@cs.rutgers.edu

**Maneesh Singh**
Verisk Analytics
maneesh.singh@verisk.com

## Abstract

Natural images may lie on a union of disjoint manifolds rather than one globally connected manifold, and this can cause several difficulties for the training of common Generative Adversarial Networks (GANs). In this work, we first show that single generator GANs are unable to correctly model a distribution supported on a disconnected manifold, and investigate how sample quality, mode dropping and local convergence are affected by this. Next, we show how using a collection of generators can address this problem, providing new insights into the success of such multi-generator GANs. Finally, we explain the serious issues caused by considering a fixed prior over the collection of generators and propose a novel approach for learning the prior and inferring the necessary number of generators without any supervision. Our proposed modifications can be applied on top of any other GAN model to enable learning of distributions supported on disconnected manifolds. We conduct several experiments to illustrate the aforementioned shortcoming of GANs, its consequences in practice, and the effectiveness of our proposed modifications in alleviating these issues.

## 1 Introduction

Consider two natural images, picture of a bird and picture of a cat for example, *can we continuously transform the bird into the cat without ever generating a picture that is not neither bird nor cat?* In other words, *is there a continuous transformation between the two that never leaves the manifold of "real looking" images?* It is often the case that real world data falls on a union of several *disjoint* manifolds and such a transformation does not exist, i.e. the real data distribution is supported on a disconnected manifold, and an effective generative model needs to be able to learn such manifolds.

Generative Adversarial Networks (GANs) [10], model the problem of finding the unknown distribution of real data as a two player game where one player, called the discriminator, tries to perfectly separate real data from the data generated by a second player, called the generator, while the second player tries to generate data that can perfectly fool the first player. Under certain conditions, Goodfellow et al. [10] proved that this process will result in a generator that generates data from the real data distribution, hence finding the unknown distribution implicitly. However, later works uncovered several shortcomings of the original formulation, mostly due to violation of one or several of its assumptions in practice [1, 2, 20, 24]. Most notably, the proof only works for when optimizing in the function space of generator and discriminator (and not in the parameter space) [10], the Jensen Shannon Divergence is maxed out when the generated and real data distributions have disjoint support

resulting in vanishing or unstable gradient [1], and finally the mode dropping problem where the generator fails to correctly capture all the modes of the data distribution, for which to the best of our knowledge there is no definitive reason yet.

One major assumption for the convergence of GANs is that the generator and discriminator both have unlimited capacity [10, 2, 24, 14], and modeling them with neural networks is then justified through the Universal Approximation Theorem. However, we should note that this theorem is only valid for continuous functions. Moreover, neural networks are far from universal approximators in practice. In fact, we often explicitly restrict neural networks through various regularizers to stabilize training and enhance generalization. Therefore, when generator and discriminator are modeled by stable regularized neural networks, they may no longer enjoy a good convergence as promised by the theory.

In this work, we focus on learning distributions with disconnected support, and show how limitations of neural networks in modeling discontinuous functions can cause difficulties in learning such distributions with GANs. We study why these difficulties arise, what consequences they have in practice, and how one can address these difficulties by using a collection of generators, providing new insights into the recent success of multi-generator models. However, while all such models consider the number of generators and the prior over them as fixed hyperparameters [3, 14, 9], we propose a novel prior learning approach and show its necessity in effectively learning a distribution with disconnected support. We would like to stress that we are not trying to achieve state of the art performance in our experiments in the present work, rather we try to illustrate an important limitation of common GAN models and the effectiveness of our proposed modifications. We summarize the contributions of this work below:

- We identify a shortcoming of GANs in modeling distributions with disconnected support, and investigate its consequences, namely mode dropping, worse sample quality, and worse local convergence (Section 2).
- We illustrate how using a collection of generators can solve this shortcoming, providing new insights into the success of multi generator GAN models in practice (Section 3).
- We show that choosing the number of generators and the probability of selecting them are important factors in correctly learning a distribution with disconnected support, and propose a novel prior learning approach to address these factors. (Section 3.1)
- Our proposed model can effectively learn distributions with disconnected supports and infer the number of necessary disjoint components through prior learning. Instead of one large neural network as the generator, it uses several smaller neural networks, making it more suitable for parallel learning and less prone to bad weight initialization. Moreover, it can be easily integrated with any GAN model to enjoy their benefits as well (Section 5).

## 2 Difficulties of Learning Disconnected Manifolds

A GAN as proposed by Goodfellow et al. [10], and most of its successors (e.g. [2, 11]) learn a continuous $G : \mathcal{Z} \to \mathcal{X}$, which receives samples from some prior $p(z)$ as input and generates real data as output. The prior $p(z)$ is often a standard multivariate normal distribution $\mathcal{N}(0, I)$ or a bounded uniform distribution $\mathcal{U}(-1, 1)$. This means that $p(z)$ is supported on a globally connected subspace of $\mathcal{Z}$. Since a continuous function always keeps the connectedness of space intact [15], the probability distribution induced by $G$ is also supported on a globally connected space. Thus $G$, a continuous function by design, can not correctly model a union of disjoint manifolds in $\mathcal{X}$. We highlight this fact in Figure 1 using an illustrative example where the support of real data is $\{+2, -2\}$. We will look at some consequences of this shortcoming in the next part of this section. For the remainder of this paper, we assume the real data is supported on a manifold $S_r$ which is a union of disjoint globally connected manifolds each denoted by $M_i$; we refer to each $M_i$ as a submanifold (note that we are overloading the topological definition of submanifolds in favor of brevity):

$$ S_r = \bigcup_{i=1}^{n_r} M_i \qquad\qquad \forall i \neq j : M_i \cap M_j = \emptyset $$

**Sample Quality.** Since GAN's generator tries to cover all submanifolds of real data with a single globally connected manifold, it will inevitably generate off real-manifold samples. Note that to avoid

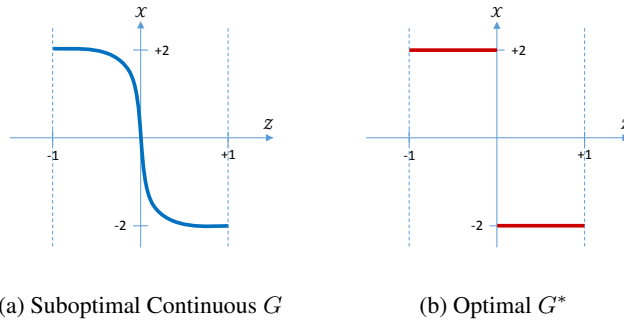

(a) Suboptimal Continuous $G$       (b) Optimal $G^*$

Figure 1: Illustrative example of continuous generator $G(z) : \mathcal{Z} \to \mathcal{X}$ with prior $z \sim \mathcal{U}(-1, 1)$, trying to capture real data coming from $p(x) = \frac{1}{2}(\delta(x - 2) + \delta(x + 2))$, a distribution supported on union of two disjoint manifolds. (a) shows an example of what a stable neural network is capable of learning for $G$ (a continuous and smooth function), (b) shows an optimal generator $G^*(z)$. Note that since $z$ is uniformly sampled, $G(z)$ is necessarily generating off manifold samples (in $[-2, 2]$) due to its continuity.

off manifold regions, one should push the generator to learn a higher frequency function, the learning of which is explicitly avoided by stable training procedures and means of regularization. Therefore the GAN model in a stable training, in addition to real looking samples, will also generate low quality off real-manifold samples. See Figure 2 for an example of this problem.

**Mode Dropping.** In this work, we use the term *mode dropping* to refer to the situation where one or several submanifolds of real data are not completely covered by the support of the generated distribution. Note that mode collapse is a special case of this definition where all but a small part of a single submanifold are dropped. When the generator can only learn a distribution with globally connected support, it has to learn a cover of the real data submanifolds, in other words, the generator can not reduce the probability density of the off real-manifold space beyond a certain value. However, the generator can try to minimize the volume of the off real-manifold space to minimize the probability of generating samples there. For example, see how in Figure 2b the learned globally connected manifold has minimum off real-manifold volume, for example it does not learn a cover that crosses the center (the same manifold is learned in 5 different runs). So, in learning the cover, there is a trade off between covering all real data submanifolds, and minimizing the volume of the off real-manifold space in the cover. This trade off means that the generator may sacrifice certain submanifolds, entirely or partially, in favor of learning a cover with less off real-manifold volume, hence mode dropping.

**Local Convergence.** Nagarajan and Kolter [21] recently proved that the training of GANs is locally convergent when generated and real data distributions are equal near the equilibrium point, and Mescheder et al. [19] showed the necessity of this condition on a prototypical example. Therefore when the generator can not learn the correct support of the real data distribution, as is in our discussion, the resulting equilibrium may not be locally convergent. In practice, this means the generator's support keeps oscillating near the data manifold.

## 3 Disconnected Manifold Learning

There are two ways to achieve disconnectedness in $\mathcal{X}$: making $\mathcal{Z}$ disconnected, or making $G : \mathcal{Z} \to \mathcal{X}$ discontinuous. The former needs considerations for how to make $\mathcal{Z}$ disconnected, for example adding discrete dimensions [6], or using a mixture of Gaussians [12]. The latter solution can be achieved by introducing a collections of independent neural networks as $G$. In this work, we investigate the latter solution since it is more suitable for parallel optimization and can be more robust to bad initialization.

We first introduce a set of generators $G_c : \mathcal{Z} \to \mathcal{X}$ instead of a single one, independently constructed on a uniform prior in the shared latent space $\mathcal{Z}$. Each generator can therefore potentially learn a separate connected manifold. However, we need to encourage these generators to each focus on a different submanifold of the real data, otherwise they may all learn a cover of the submanifolds and

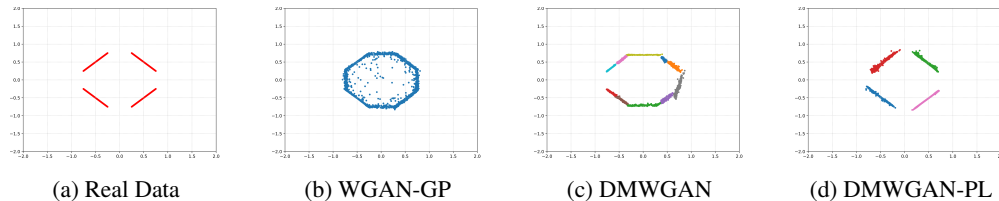

| (a) Real Data | (b) WGAN-GP | (c) DMWGAN | (d) DMWGAN-PL |

Figure 2: Comparing Wasserstein GAN (WGAN) and its Disconnected Manifold version with and without prior learning (DMWGAN-PL, DMWGAN) on disjoint line segments dataset when $n_g = 10$. Different colors indicate samples from different generators. Notice how WGAN-GP fails to capture the disconnected manifold of real data, learning a globally connected cover instead, and thus generating off real-manifold samples. DMWGAN also fails due to incorrect number of generators. In contrast, DMWGAN-PL is able to infer the necessary number of disjoint components without any supervision and learn the correct disconnected manifold of real data. Each figure shows 10K samples from the respective model. We train each model 5 times, the results shown are consistent across different runs.

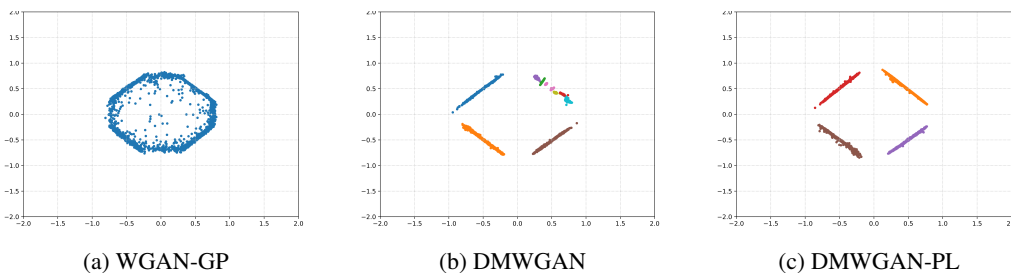

| (a) WGAN-GP | (b) DMWGAN | (c) DMWGAN-PL |

Figure 3: Comparing WGAN-GP, DMWGAN and DMWGAN-PL convergence on unbalanced disjoint line segments dataset when $n_g = 10$. The real data is the same line segments as in Figure 2, except the top right line segment has higher probability. Different colors indicate samples from different generators. Notice how DMWGAN-PL (c) has vanished the contribution of redundant generators wihtout any supervision. Each figure shows 10K samples from the respective model. We train each model 5 times, the results shown are consistent across different runs.

experience the same issues of a single generator GAN. Intuitively, we want the samples generated by each generator to be perfectly unique to that generator, in other words, each sample should be a perfect indicator of which generator it came from. Naturally, we can achieve this by maximizing the mutual information $\mathcal{I}(c; x)$, where $c$ is generator id and $x$ is generated sample. As suggested by Chen et al. [6], we can implement this by maximizing a lower bound on mutual information between generator ids and generated images:

$$
\begin{aligned}
\mathcal{I}(c; x) &= H(c) - H(c|x) \\
&= H(c) + \mathbb{E}_{c \sim p(c), x \sim p_g(x|c)} \left[ \mathbb{E}_{c' \sim p(c'|x)} \left[ \ln p(c'|x) \right] \right] \\
&= H(c) + \mathbb{E}_{x \sim p_g(x)} \left[ \mathcal{KL}(p(c|x) \| q(c|x)) \right] + \mathbb{E}_{c \sim p(c), x \sim p_g(x|c), c' \sim p(c'|x)} \left[ \ln q(c'|x) \right] \\
&\geq H(c) + \mathbb{E}_{c \sim p(c), x \sim p_g(x|c), c' \sim p(c'|x)} \left[ \ln q(c'|x) \right] \\
&= H(c) + \mathbb{E}_{c \sim p(c), x \sim p_g(x|c)} \left[ \ln q(c|x) \right]
\end{aligned}
$$

where $q(c|x)$ is the distribution approximating $p(c|x)$, $p_g(x|c)$ is induced by each generator $G_c$, $\mathcal{KL}$ is the Kullback Leibler divergence, and the last equality is a consequence of Lemma 5.1 in [6]. Therefore, by modeling $q(c|x)$ with a neural network $Q(x; \gamma)$, the encoder network, maximizing $\mathcal{I}(c; x)$ boils down to minimizing a cross entropy loss:

$$
L_c = -\mathbb{E}_{c \sim p(c), x \sim p_g(x|c)} \left[ \ln q(c|x) \right] \tag{1}
$$

Utilizing the Wasserstein GAN [2] objectives, discriminator (critic) and generator maximize the following, where $D(x; w) : \mathcal{X} \to \mathbb{R}$ is the critic function:

$$V_d = \mathbb{E}_{x \sim p_r(x)} \left[ D(x; w) \right] - \mathbb{E}_{c \sim p(c), x \sim p_g(x|c)} \left[ D(x; w) \right] \qquad (2)$$

$$V_g = \mathbb{E}_{c \sim p(c), x \sim p_g(x|c)} \left[ D(x; w) \right] - \lambda L_c \qquad (3)$$

We call this model Disconnected Manifold Learning WGAN (DMWGAN) in our experiments. We can similarly apply our modifications to the original GAN [10] to construct DMGAN. We add the single sided version of penalty gradient regularizer [11] to the discriminator/critic objectives of both models and all baselines. See Appendix A for details of our algorithm and the DMGAN objectives. See Appendix F for more details and experiments on the importance of the mutual information term.

The original convergence theorems of Goodfellow et al. [10] and Arjovsky et al. [2] holds for the proposed DM versions respectively, because all our modifications concern the internal structure of the generator, and can be absorbed into the unlimited capacity assumption. More concretely, all generators together can be viewed as a unified generator where $p(c)p_g(x|c)$ becomes the generator probability, and $L_c$ can be considered as a constraint on the generator function space incorporated using a Lagrange multiplier. While most multi-generator models consider $p(c)$ as a uniform distribution over generators, this naive choice of prior can cause certain difficulties in learning a disconnected support. We will discuss this point, and also introduce and motivate the metrics we use for evaluations, in the next two subsections.

## 3.1   Learning the Generator's Prior

In practice, we can not assume that the true number of submanifolds in real data is known a priori. So let us consider two cases regarding the number of generators $n_g$, compared to the true number of submanifolds in data $n_r$, under a fixed uniform prior $p(c)$. If $n_g < n_r$ then some generators have to cover several submanifolds of the real data, thus partially experiencing the same issues discussed in Section 2. If $n_g > n_r$, then some generators have to share one real submanifold, and since we are forcing the generators to maintain disjoint supports, this results in partially covered real submanifolds, causing mode dropping. See Figures 2c and 3b for examples of this issue. Note that an effective solution to the latter problem reduces the former problem into a trade off: the more the generators, the better the cover. We can address the latter problem by learning the prior $p(c)$ such that it vanishes the contribution of redundant generators. Even when $n_g = n_r$, what if the distribution of data over submanifolds are not uniform? Since we are forcing each generator to learn a different submanifold, a uniform prior over the generators would result in a suboptimal distribution. This issue further shows the necessity of learning the prior over generators.

We are interested in finding the best prior $p(c)$ over generators. Notice that $q(c|x)$ is implicitly learning the probability of $x \in \mathcal{X}$ belonging to each generator $G_c$, hence $q(c|x)$ is approximating the true posterior $p(c|x)$. We can take an EM approach to learning the prior: the expected value of $q(c|x)$ over the real data distribution gives us an approximation of $p(c)$ (E step), which we can use to train the DMGAN model (M step). Instead of using empirical average to learn $p(c)$ directly, we learn it with a model $r(c; \zeta)$, which is a softmax function over parameters $\{\zeta_i\}_{i=1}^{n_g}$ corresponding to each generator. This enables us to control the learning of $p(c)$, the advantage of which we will discuss shortly. We train $r(c)$ by minimizing the cross entropy as follows:

$$H(p(c), r(c)) = -\mathbb{E}_{c \sim p(c)} \left[ \log r(c) \right] = -\mathbb{E}_{x \sim p_r(x), c \sim p(c|x)} \left[ \log r(c) \right] = \mathbb{E}_{x \sim p_r(x)} \left[ H(p(c|x), r(c)) \right]$$

Where $H(p(c|x), r(c))$ is the cross entropy between model distribution $r(c)$ and true posterior $p(c|x)$ which we approximate by $q(c|x)$. However, learning the prior from the start, when the generators are still mostly random, may prevent most generators from learning by vanishing their probability too early. To avoid this problem, we add an entropy regularizer and decay its weight $\lambda''$ with time to gradually shift the prior $r(c)$ away from uniform distribution. Thus the final loss for training $r(c)$ becomes:

$$L_{prior} = \mathbb{E}_{x \sim p_r(x)} \left[ H(q(c|x), r(c)) \right] - \alpha^t \lambda'' H(r(c)) \qquad (4)$$

Where $H(r(c))$ is the entropy of model distribution, $\alpha$ is the decay rate, and $t$ is training timestep. The model is not very sensitive to $\lambda''$ and $\alpha$, any combination that insures a smooth transition away from uniform distribution is valid. We call this augmented model Disconnected Manifold Learning GAN with Prior Learning (DMGAN-PL) in our experiments. See Figures 2 and 3 for examples showing the advantage of learning the prior.

## 3.2 Choice of Metrics

We require metrics that can assess inter-mode variation, intra-mode variation and sample quality. The common metric, Inception Score [23], has several drawbacks [4, 18], most notably it is indifferent to intra-class variations and favors generators that achieve close to uniform distribution over classes of data. Instead, we consider more direct metrics together with FID score [13] for natural images.

For inter mode variation, we use the Jensen Shannon Divergence (JSD) between the class distribution of a pre-trained classifier over real data and generator's data. This can directly tell us how well the distribution over classes are captured. JSD is favorable to KL due to being bounded and symmetric. For intra mode variation, we define mean square geodesic distance (MSD): the average squared geodesic distance between pairs of samples classified into each class. To compute the geodesic distance, Euclidean distance is used in a small neighborhood of each sample to construct the Isomap graph [26] over which a shortest path distance is calculated. This shortest path distance is an approximation to the geodesic distance on the true image manifold [25]. Note that average square distance, for Euclidean distance, is equal to twice the trace of the Covariance matrix, i.e. sum of the eigenvalues of covariance matrix, and therefore can be an indicator of the variance within each class:

$$\mathbb{E}_{x,y}\left[||x - y||^2\right] = 2\mathbb{E}_x\left[x^T x\right] - 2\mathbb{E}_x\left[x\right]^T \mathbb{E}_x\left[x\right] = 2Tr(Cov(x))$$

In our experiments, we choose the smallest $k$ for which the constructed k nearest neighbors graph (Isomap) is connected in order to have a better approximation of the geodesic distance ($k = 18$).

Another concept we would like to evaluate is sample quality. Given a pretrained classifier with small test error, samples that are classified with high confidence can be reasonably considered good quality samples. We plot the ratio of samples classified with confidence greater than a threshold, versus the confidence threshold, as a measure of sample quality: the more off real-manifold samples, the lower the resulting curve. Note that the results from this plot are exclusively indicative of sample quality and should be considered in conjunction with the aforementioned metrics.

What if the generative model memorizes the dataset that it is trained on? Such a model would score perfectly on all our metrics, while providing no generalization at all. First, note that a single generator GAN model can not memorize the dataset because it can not learn a distribution supported on $N$ disjoint components as discussed in Section 2. Second, while our modifications introduces disconnnectedness to GANs, the number of generators we use in our proposed modifications are in the order of data submanifolds which is several orders of magnitude less than common dataset sizes. Note that if we were to assign one unique point of the $\mathcal{Z}$ space to each dataset sample, then a neural network could learn to memorize the dataset by mapping each selected $z \in \mathcal{Z}$ to its corresponding real sample (we have introduced $N$ disjoint component in $\mathcal{Z}$ space in this case), however this is not how GANs are modeled. Therefore, the memorization issue is not of concern for common GANs and our proposed models (note that this argument is addressing the very narrow case of dataset memorization, not over-fitting in general).

## 4 Related Works

Several recent works have directly targeted the mode collapse problem by introducing a network $F : \mathcal{X} \rightarrow \mathcal{Z}$ that is trained to map back the data into the latent space prior $p(z)$. It can therefore provide a learning signal if the generated data has collapsed. ALI [8] and BiGAN [7] consider pairs of data and corresponding latent variable $(x, z)$, and construct their discriminator to distinguish such pairs of real and generated data. VEEGAN [24] uses the same discriminator, but also adds an explicit reconstruction loss $\mathbb{E}_{z \sim p(z)}\left[||z - F_\theta(G_\gamma(z))||_2^2\right]$. The main advantage of these models is to prevent loss of information by the generator (mapping several $z \in \mathcal{Z}$ to a single $x \in \mathcal{X}$). However, in case of distributions with disconnected support, these models do not provide much advantage over common GANs and suffer from the same issues we discussed in Section 2 due to having a single generator.

Another set of recent works have proposed using multiple generators in GANs in order to improve their convergence. MIX+GAN [3] proposes using a collection of generators based on the well-known advantage of learning a mixed strategy versus a pure strategy in game theory. MGAN [14] similarly uses a collection of $k$ generators in order to model a mixture distribution, and train them together with a k-class classifier to encourage them to each capture a different component of the real mixture distribution. MAD-GAN [9], also uses $k$ generators, together with a $k + 1$-class discriminator which is trained to correctly classify samples from each generator and true data (hence a $k + 1$ classifier),

| Model | JSD MNIST $\times 10^{-2}$ | JSD Face-Bed $\times 10^{-4}$ | FID Face-Bed |
|---|---|---|---|
| WGAN-GP | 0.13 std 0.05 | 0.23 std 0.15 | 8.30 std 0.27 |
| MIX+GAN | 0.17 std 0.08 | 0.83 std 0.57 | 8.02 std 0.14 |
| DMWGAN | 0.23 std 0.06 | 0.46 std 0.25 | 7.96 std 0.08 |
| DMWGAN-PL | 0.06 std 0.02 | 0.10 std 0.05 | 7.67 std 0.16 |

Table 1: Inter-class variation measured by Jensen Shannon Divergence (JSD) with true class distribution for MNIST and Face-Bedroom dataset, and FID score for Face-Bedroom (smaller is better). We run each model 5 times with random initialization, and report average values with one standard deviation interval

in order to increase the diversity of generated images. While these models provide reasons for why multiple generators can model mixture distributions and achieve more diversity, they do not address why single generator GANs fail to do so. In this work, we explain why it is the disconnectedness of the support that single generator GANs are unable to learn, not the fact that real data comes from a mixture distribution. Moreover, all of these works use a fixed number of generators and do not have any prior learning, which can cause serious problems in learning of distributions with disconnected support as we discussed in Section 3.1 (see Figures 2c and 3b for examples of this issue).

Finally, several works have targeted the problem of learning the correct manifold of data. MDGAN [5], uses a two step approach to closely capture the manifold of real data. They first approximate the data manifold by learning a transformation from encoded real images into real looking images, and then train a single generator GAN to generate images similar to the transformed encoded images of previous step. However, MDGAN can not model distributions with disconnected supports. InfoGAN [6] introduces auxiliary dimensions to the latent space $\mathcal{Z}$, and maximizes the mutual information between these extra dimensions and generated images in order to learn disentangled representations in the latent space. DeLiGAN [12] uses a fixed mixture of Gaussians as its latent prior, and does not have any mechanisms to encourage diversity. While InfoGAN and DeLiGAN can generate disconnected manifolds, they both assume a fixed number of discreet components equal to the number of underlying classes and have no prior learning over these components, thus suffering from the issues discussed in Section 3.1. Also, neither of these works discusses the incapability of single generator GANs to learn disconnected manifolds and its consequences.

## 5 Experiments

In this section we present several experiments to investigate the issues and proposed solutions mentioned in Sections 2 and 3 respectively. The same network architecture is used for the discriminator and generator networks of all models under comparison, except we use $\frac{1}{4}$ number of filters in each layer of multi-generator models compared to the single generator models, to control the effect of complexity. In all experiments, we train each model for a total of 200 epochs with a five to one update ratio between discriminator and generator. $Q$, the encoder network, is built on top of discriminator's last hidden layer, and is trained simultaneously with generators. Each data batch is constructed by first selecting 32 generators according to the prior $r(c; \zeta)$, and then sampling each one using $z \sim \mathcal{U}(-1, 1)$. See Appendix B for details of our networks and the hyperparameters.

**Disjoint line segments.** This dataset is constructed by sampling data with uniform distribution over four disjoint line segments to achieve a distribution supported on a union of disjoint low-dimensional manifolds. See Figure 2 for the results of experiments on this dataset. In Figure 3, an unbalanced version of this dataset is used, where 0.7 probability is placed on the top right line segment, and the other segments have 0.1 probability each. The generator and discriminator are both MLPs with two hidden layers, and 10 generators are used for multi-generator models. We choose WGAN-GP as the state of the art GAN model in these experiments (we observed similar or worse convergence with other flavors of single generator GANs). MGAN achieves similar results to DMWGAN.

**MNIST dataset.** MNIST [16] is particularly suitable since samples with different class labels can be reasonably interpreted as lying on disjoint manifolds (with minor exceptions like certain 4s and 9s). The generator and discriminator are DCGAN like networks [22] with three convolution layers. Figure 4 shows the mean squared geodesic distance (MSD) and Table 1 reports the corresponding

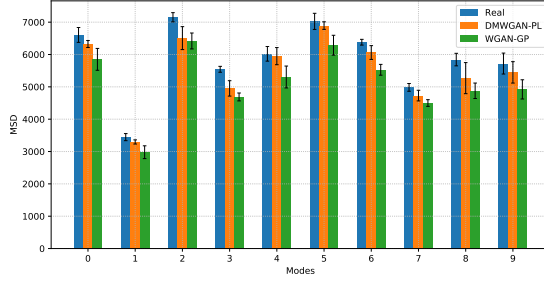

(a) Intra-class variation MNIST

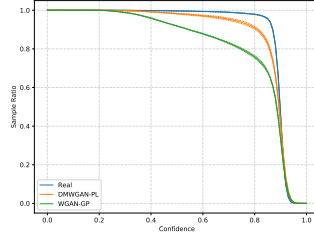
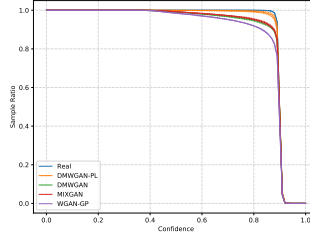

(b) Sample quality MNIST          (c) Sample quality Face-Bed

Figure 4: (a) Shows intra-class variation in MNIST. Bars show the mean square distance (MSD) within each class of the dataset. On average, DMGAN-PL outperforms WGAN-GP in capturing intra class variation, as measured by MSD, with larger significance on certain classes. (b) Shows the sample quality in MNIST experiment. (c) Shows sample quality in Face-Bed experiment. Notice how DMWGAN-PL outperforms other models due to fewer off real-manifold samples. We run each model 5 times with random initialization, and report average values with one standard deviation intervals in both figures. 10K samples are used for metric evaluations.

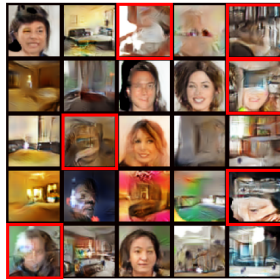
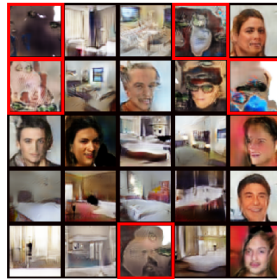
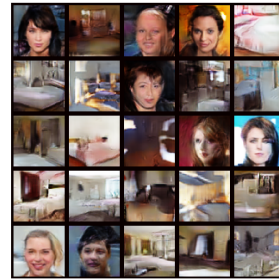

(a) WGAN-GP                (b) DMWGAN                (c) DMWGAN-PL

Figure 5: Samples randomly generated by GAN models trained on Face-Bed dataset. Notice how WGAN-GP generates combined face-bedroom images (red boxes) in addition to faces and bedrooms, due to learning a connected cover of the real data support. DMWGAN does not generate such samples, however it generates completely off manifold samples (red boxes) due to having redundant generators and a fixed prior. DMWGAN-PL is able to correctly learn the disconnected support of real data. The samples and trained models are not cherry picked.

divergences in order to compare their inter mode variation. 20 generators are used for multi-generator models. See Appendix C for experiments using modified GAN objective. Results demonstrate the advantage of adding our proposed modification on both GAN and WGAN. See Appendix D for qualitative results.

**Face-Bed dataset.** We combine 20K face images from CelebA dataset [17] and 20K bedroom images from LSUN Bedrooms dataset [27] to construct a natural image dataset supported on a disconnected manifold. We center crop and resize images to $64 \times 64$. 5 generators are used for multi-generator

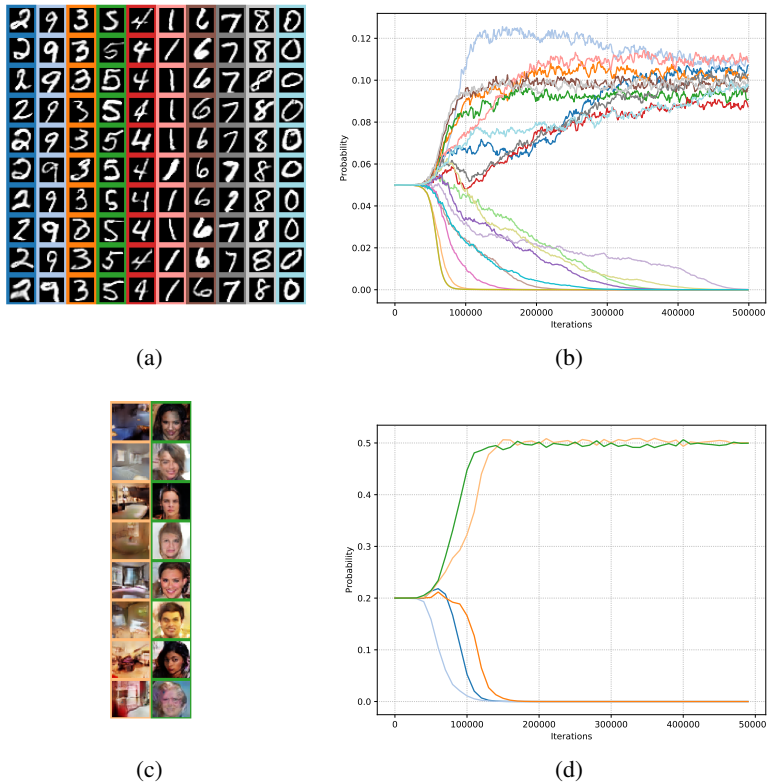

<div align="center">(a)</div>
<div align="center">(b)</div>

<div align="center">(c)</div>
<div align="center">(d)</div>

Figure 6: DMWGAN-PL prior learning during training on MNIST with 20 generators (a,b) and on Face-Bed with 5 generators (c, d). (a, c) show samples from top generators with prior greater than $0.05$ and $0.2$ respectively. (b, d) show the probability of selecting each generator $r(c; \zeta)$ during training, each color denotes a different generator. The color identifying each generator in (b) and the border color of each image in (a) are corresponding, similarly for (d) and (c). Notice how prior learning has correctly learned probability of selecting each generators and dropped out redundant generators without any supervision.

models. Figures 4c, 5 and Table 1 show the results of this experiment. See Appendix E for more qualitative results.

## 6 Conclusion and Future Works

In this work we showed why the single generator GANs can not correctly learn distributions supported on disconnected manifolds, what consequences this shortcoming has in practice, and how multi-generator GANs can effectively address these issues. Moreover, we showed the importance of learning a prior over the generators rather than using a fixed prior in multi-generator models. However, it is important to highlight that throughout this work we assumed the disconnectedness of the real data support. Verifying this assumption in major datasets, and studying the topological properties of these datasets in general, are interesting future works. Extending the prior learning to other methods, such as learning a prior over shape of $\mathcal{Z}$ space, and also investigating the effects of adding diversity to discriminator as well as the generators, also remain as exciting future paths for research.

## Acknowledgement

This work was supported by Verisk Analytics and NSF-USA award number 1409683.

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
