[Supplementary Material]

# Disconnected Manifold Learning for Generative Adversarial Networks: Appendix

## Appendix A   Algorithm

---

**Algorithm 1** Disconnected Manifold Learning WGAN with Prior Learning (DMWGAN-PL). Replace $V_d$ and $V_g$ according to Eq 5 and Eq 6 in lines 7 and 15 for the Modified GAN version (DMGAN-PL).

---

**Precondition:** $p(z)$ prior on $\mathcal{Z}$, $m$ batch size, $k$ number of discriminator updates, $n_g$ number of generators, $\lambda = 1$, $\lambda' = 10$ and $\lambda'' = 1000$ are weight coefficients, $\alpha = 0.999$ is decay rate, and $t = 0$

1: **repeat**
2:     **for** $j \in \{1 \ldots k\}$ **do**
3:         $\{x^i\}_{i=1}^m \sim p_r(x)$                                     $\triangleright$ A batch from real data
4:         $\{z^i\}_{i=1}^m \sim p(z)$                                        $\triangleright$ A batch from $\mathcal{Z}$ prior
5:         $\{c^i\}_{i=1}^m \sim r(c; \zeta)$                                $\triangleright$ A batch from generator's prior
6:         $\{x_g^i\}_{i=1}^m \leftarrow G(z^i; \theta_{c^i})$                    $\triangleright$ Generate batch using selected generators
7:         $g_w \leftarrow \nabla_w \frac{1}{m} \sum_i \left[ D(x^i; w) - D(x_g^i; w) + \lambda' V_{regul} \right]$
8:         $w \leftarrow \text{Adam}(w, g_w)$                                $\triangleright$ Maximize $V_d$ wrt. $w$
9:     **end for**
10:    $\{x^i\}_{i=1}^m \sim p_r(x)$
11:    $\{z^i\}_{i=1}^m \sim p(z)$
12:    $\{c^i\}_{i=1}^m \sim r(c; \zeta)$
13:    $\{x_g^i\}_{i=1}^m \leftarrow G(z^i; \theta_{c^i})$
14:    **for** $j \in \{1 \ldots n_g\}$ **do**
15:         $g_{\theta_j} \leftarrow \nabla_{\theta_j} \frac{1}{m} \sum_i \left[ D(x_g^i; w) - \lambda \ln Q(x_g^i; \gamma) \right]$         $\triangleright$ $\theta_j$ is short for $\theta_{c^j}$
16:         $\theta_j \leftarrow \text{Adam}(g_{\theta_j}, \theta_j)$                              $\triangleright$ Maximize $V_g$ wrt. $\theta$
17:    **end for**
18:    $g_\gamma \leftarrow \nabla_\gamma \frac{1}{m} \sum_i \ln Q(x_g^i; \gamma)$
19:    $\gamma \leftarrow \text{Adam}(g_\gamma, \gamma)$                                  $\triangleright$ Minimize $L_c$ wrt. $\gamma$
20:    $g_\zeta \leftarrow \nabla_\zeta \frac{1}{m} \sum_i \left[ H(Q(x^i; \gamma), r(c; \zeta)) \right] - \alpha^t \lambda'' H(r(c; \zeta))$
21:    $\zeta \leftarrow \text{Adam}(g_\zeta, \zeta)$                                  $\triangleright$ Minimize $L_{prior}$ wrt. $\zeta$
22:    $t \leftarrow t + 1$
23: **until** convergence.

---

Utilizing the modified GAN objectives, discriminator and generator maximize the following:

$$V_d = \mathbb{E}_{x \sim p_r(x)} \left[ \ln D(x; w) \right] + \mathbb{E}_{c \sim p(c), x \sim p_g(x|c)} \left[ \ln(1 - D(x; w)) \right] \tag{5}$$

$$V_g = \mathbb{E}_{c \sim p(c), x \sim p_g(x|c)} \left[ \ln D(x; w) \right] - \lambda L_c \tag{6}$$

Where $p_g(x|c)$ is the distribution induced by the $c$th generator modeled by a neural network $G(z; \theta_c)$, and $D(x; w) : \mathcal{X} \to [0, 1]$ is the discriminator function. We add the single sided version of penalty gradient regularizer with a weight $\lambda'$ to the discriminator/critic objectives of both versions of DMGAN and all our baselines:

$$V_{regul} = -\mathbb{E}_{x \sim p_l(x)} \left[ \max(||\nabla D(x)||_2 - 1, 0)^2 \right] \tag{7}$$

Where $p_l(x)$ is induced by uniformly sampling from the line connecting a sample from $p_r(x)$ and a sample from $p_g(x)$.

# Appendix B   Network Architecture

Table 2: Single Generator Models MNIST

| Operation | Kernel | Strides | Feature Maps | BN | Activation |
|---|---|---|---|---|---|
| G(z): $z \sim Uniform[-1,1]$ | | | 100 | | |
| Fully Connected | | | $4 \times 4 \times 128$ | - | ReLU |
| Nearest Up Sample | | | $8 \times 8 \times 128$ | - | |
| Convolution | $5 \times 5$ | $1 \times 1$ | $8 \times 8 \times 64$ | - | ReLU |
| Nearest Up Sample | | | $16 \times 16 \times 64$ | - | |
| Convolution | $5 \times 5$ | $1 \times 1$ | $16 \times 16 \times 32$ | - | ReLU |
| Nearest Up Sample | | | $28 \times 28 \times 32$ | - | |
| Convolution | $5 \times 5$ | $1 \times 1$ | $28 \times 28 \times 1$ | - | Tanh |
| D(x) | | | $28 \times 28 \times 1$ | | |
| Convolution | $5 \times 5$ | $2 \times 2$ | $14 \times 14 \times 32$ | - | Leaky ReLU |
| Convolution | $5 \times 5$ | $2 \times 2$ | $7 \times 7 \times 64$ | - | Leaky ReLU |
| Convolution | $5 \times 5$ | $2 \times 2$ | $4 \times 4 \times 128$ | - | Leaky ReLU |
| Fully Connected | | | 1 | - | Sigmoid |
| Batch size | 32 | | | | |
| Leaky ReLU slope | 0.2 | | | | |
| Gradient Penalty weight | 10 | | | | |
| Learning Rate | 0.0002 | | | | |
| Optimizer | Adam | $\beta_1 = 0.5$ | $\beta_2 = 0.5$ | | |
| Weight Bias init | Xavier, 0 | | | | |

Table 3: Multiple Generator Models MNIST

| Operation | Kernel | Strides | Feature Maps | BN | Activation | Shared? |
|---|---|---|---|---|---|---|
| G(z): $z \sim Uniform[-1,1]$ | | | 100 | | | N |
| Fully Connected | | | $4 \times 4 \times 32$ | - | ReLU | N |
| Nearest Up Sample | | | $8 \times 8 \times 32$ | - | | N |
| Convolution | $5 \times 5$ | $1 \times 1$ | $8 \times 8 \times 16$ | - | ReLU | N |
| Nearest Up Sample | | | $16 \times 16 \times 16$ | - | | N |
| Convolution | $5 \times 5$ | $1 \times 1$ | $16 \times 16 \times 8$ | - | ReLU | N |
| Nearest Up Sample | | | $28 \times 28 \times 8$ | - | | N |
| Convolution | $5 \times 5$ | $1 \times 1$ | $28 \times 28 \times 1$ | - | Tanh | N |
| Q(x), D(x) | | | $28 \times 28 \times 1$ | | | |
| Convolution | $5 \times 5$ | $2 \times 2$ | $14 \times 14 \times 32$ | - | Leaky ReLU | Y |
| Convolution | $5 \times 5$ | $2 \times 2$ | $7 \times 7 \times 64$ | - | Leaky ReLU | Y |
| Convolution | $5 \times 5$ | $2 \times 2$ | $4 \times 4 \times 128$ | - | Leaky ReLU | Y |
| D Fully Connected | | | 1 | - | Sigmoid | N |
| Q Convolution | $5 \times 5$ | $2 \times 2$ | $4 \times 4 \times 128$ | Y | Leaky ReLU | N |
| Q Fully Connected | | | $n_g$ | - | Softmax | N |
| Batch size | 32 | | | | | |
| Leaky ReLU slope | 0.2 | | | | | |
| Gradient Penalty weight | 10 | | | | | |
| Learning Rate | 0.0002 | | | | | |
| Optimizer | Adam | $\beta_1 = 0.5$ | $\beta_2 = 0.5$ | | | |
| Weight Bias init | Xavier, 0 | | | | | |

The pre-trained classifier used for metrics is the ALL-CNN-B model from Springenberg et al. [2] trained to test accuracy 0.998 on MNIST and 1.000 on Face-Bed.

Table 4: Single Generator Models Face-Bed

| Operation | Kernel | Strides | Feature Maps | BN | Activation |
|---|---|---|---|---|---|
| G(z): $z \sim Uniform[-1,1]$ | | | 100 | | |
| Fully Connected | | | $8 \times 8 \times 512$ | - | ReLU |
| Nearest Up Sample | | | $16 \times 16 \times 512$ | - | |
| Convolution | $5 \times 5$ | $1 \times 1$ | $16 \times 16 \times 256$ | - | ReLU |
| Nearest Up Sample | | | $32 \times 32 \times 256$ | - | |
| Convolution | $5 \times 5$ | $1 \times 1$ | $32 \times 32 \times 128$ | - | ReLU |
| Nearest Up Sample | | | $64 \times 64 \times 128$ | - | |
| Convolution | $5 \times 5$ | $1 \times 1$ | $64 \times 64 \times 3$ | - | Tanh |
| D(x) | | | $32 \times 32 \times 3$ | | |
| Convolution | $5 \times 5$ | $2 \times 2$ | $32 \times 32 \times 128$ | - | Leaky ReLU |
| Convolution | $5 \times 5$ | $2 \times 2$ | $16 \times 16 \times 256$ | - | Leaky ReLU |
| Convolution | $5 \times 5$ | $2 \times 2$ | $8 \times 8 \times 512$ | - | Leaky ReLU |
| Fully Connected | | | 1 | - | Sigmoid |
| Batch size | 32 | | | | |
| Leaky ReLU slope | 0.2 | | | | |
| Gradient Penalty weight | 10 | | | | |
| Learning Rate | 0.0002 | | | | |
| Optimizer | Adam | $\beta_1 = 0.5$ | $\beta_2 = 0.5$ | | |
| Weight Bias init | Xavier, 0 | | | | |

Table 5: Multiple Generator Models Face-Bed

| Operation | Kernel | Strides | Feature Maps | BN | Activation | Shared? |
|---|---|---|---|---|---|---|
| G(z): $z \sim Uniform[-1,1]$ | | | 100 | | | N |
| Fully Connected | | | $8 \times 8 \times 128$ | - | ReLU | N |
| Nearest Up Sample | | | $16 \times 16 \times 128$ | - | | N |
| Convolution | $5 \times 5$ | $1 \times 1$ | $16 \times 16 \times 64$ | - | ReLU | N |
| Nearest Up Sample | | | $32 \times 32 \times 64$ | - | | N |
| Convolution | $5 \times 5$ | $1 \times 1$ | $32 \times 32 \times 32$ | - | ReLU | N |
| Nearest Up Sample | | | $64 \times 64 \times 32$ | - | | N |
| Convolution | $5 \times 5$ | $1 \times 1$ | $64 \times 64 \times 3$ | - | Tanh | N |
| Q(x), D(x) | | | $64 \times 64 \times 3$ | | | |
| Convolution | $5 \times 5$ | $2 \times 2$ | $32 \times 32 \times 128$ | - | Leaky ReLU | Y |
| Convolution | $5 \times 5$ | $2 \times 2$ | $16 \times 16 \times 256$ | - | Leaky ReLU | Y |
| Convolution | $5 \times 5$ | $2 \times 2$ | $8 \times 8 \times 512$ | - | Leaky ReLU | Y |
| D Fully Connected | | | 1 | - | Sigmoid | N |
| Q Convolution | $5 \times 5$ | $2 \times 2$ | $8 \times 8 \times 512$ | Y | Leaky ReLU | N |
| Q Fully Connected | | | $n_g$ | - | Softmax | N |
| Batch size | 32 | | | | | |
| Leaky ReLU slope | 0.2 | | | | | |
| Gradient Penalty weight | 10 | | | | | |
| Learning Rate | 0.0002 | | | | | |
| Optimizer | Adam | $\beta_1 = 0.5$ | $\beta_2 = 0.5$ | | | |
| Weight Bias init | Xavier, 0 | | | | | |

Table 6: Single Generator Models Disjoint Line Segments

| Operation | Kernel | Strides | Feature Maps | BN | Activation |
|---|---|---|---|---|---|
| G(z): $z \sim Uniform[-1,1]$ | | | 100 | | |
| Fully Connected | | | 128 | - | ReLU |
| Fully Connected | | | 64 | - | ReLU |
| Fully Connected | | | 2 | - | |
| D(x) | | | 2 | | |
| Fully Connected | | | 64 | - | Leaky ReLU |
| Fully Connected | | | 128 | - | Leaky ReLU |
| Fully Connected | | | 1 | - | Sigmoid |
| Batch size | 32 | | | | |
| Leaky ReLU slope | 0.2 | | | | |
| Gradient Penalty weight | 10 | | | | |
| Learning Rate | 0.0002 | | | | |
| Optimizer | Adam | $\beta_1 = 0.5$ | $\beta_2 = 0.5$ | | |
| Weight Bias init | Xavier, 0 | | | | |

Table 7: Multiple Generator Models Disjoint Line Segments

| Operation | Kernel | Strides | Feature Maps | BN | Activation | Shared? |
|---|---|---|---|---|---|---|
| G(z): $z \sim Uniform[-1,1]$ | | | 100 | | | N |
| Fully Connected | | | 32 | - | ReLU | N |
| Fully Connected | | | 16 | - | ReLU | N |
| Fully Connected | | | 2 | - | ReLU | N |
| Q(x), D(x) | | | 2 | | | |
| Fully Connected | | | 64 | - | Leaky ReLU | Y |
| Fully Connected | | | 128 | - | Leaky ReLU | Y |
| D Fully Connected | | | 1 | - | Sigmoid | N |
| Q Fully Connected | | | 128 | Y | Sigmoid | N |
| Q Fully Connected | | | $n_g$ | - | Softmax | N |
| Batch size | 32 | | | | | |
| Leaky ReLU slope | 0.2 | | | | | |
| Gradient Penalty weight | 10 | | | | | |
| Learning Rate | 0.0002 | | | | | |
| Optimizer | Adam | $\beta_1 = 0.5$ | $\beta_2 = 0.5$ | | | |
| Weight Bias init | Xavier, 0 | | | | | |

# Appendix C    DMGAN on MNIST dataset

Table 8: Inter-class variation in MNIST under GAN modified objective with gradient penalty [1]. We show the Kullback Leibler Divergence, $\text{KL}(p \parallel g)$, its inverse $\text{KL}(g \parallel p)$, and the Jensen Shannon Divergence, $\text{JSD}(p \parallel g)$ for true data and each model, where $p$ and $g$ are the distribution of images over classes using ground truth labels and a pretrained classifier respectively. Each row corresponds to the $g$ retrieved from the respective model. The results show that DMGAN-PL captures the inter class variation better that GAN. We run each model 5 times with random initialization, and report average divergences with one standard deviation interval

| Model | KL | Reverse KL | JSD |
|---|---|---|---|
| Real | 0.0005 std 0.0002 | 0.0005 std 0.0002 | 0.0001 std 0.0000 |
| GAN-GP | 0.0210 std 0.0029 | 0.0205 std 0.0030 | 0.0052 std 0.0007 |
| DMGAN-PL | 0.0020 std 0.0009 | 0.0020 std 0.0009 | 0.0005 std 0.0002 |

(a) Intra-class variation

(b) Sample quality

Figure 7: MNIST experiment under GAN modified objective with gradient penalty. (a) Shows intra-class variation. Bars show the mean square distance (MSD) within each class of the dataset for real data, DMGAN-PL model, and GAN model, using the pretrained classifier to determine classes. On average, DMGAN-PL outperforms GAN in capturing intra class variation, as measured by MSD, with larger significance on certain classes. (b) Shows the ratio of samples classified with high confidence by the pretrained classifier, a measure of sample quality. We run each model 5 times with random initialization, and report average values with one standard deviation intervals in both figures. 10K samples are used for metric evaluations.

# Appendix D    MNIST qualitative results

(a) Real Data        (b) DMWGAN-PL        (c) WGAN-GP

Figure 8: Samples randomly generated from (a) MNIST dataset, (b) DMWGAN-PL model, and (c) WGAN-GP model. Notice the better variation in DMWGAN-PL and the off manifold samples in WGAN-GP (in 6th column, the 2nd and 4th row position for example). The samples and trained models are not cherry picked.

# Appendix E   Face-Bed qualitative results

(a) Real Data

(b) WGAN-GP

(c) MIX+GAN

(d) DMWGAN (MGAN)

(e) DMWGAN-PL

Figure 9: Samples randomly generated from each model. Notice how models without prior learning generate off real-manifold images, that is they generate samples that are combination of bedrooms and faces (hence neither face nor bedroom), in addition to correct face and bedroom images. The samples and trained models are not cherry picked.

(a) MIX+GAN       (b) DMWGAN (MGAN)       (c) DMWGAN-PL

Figure 10: Samples randomly generated from the generators of each model (each column corresponds to a different generator). Notice how MIX+GAN and DMWGAN both have good generators together with very low quality generators due to sharing the two real image manifolds among all their 5 generators (they uniformly generate samples from their generators). However, DMWGAN-PL has effectively dropped the training of redundant generators, only focusing on the two necessary ones, without any supervision (only selects samples from the last two). Samples and models are not cherry picked.

# Appendix F    Importance of Mutual Information

As discussed in Section 3, maximizing mutual information (MI) between generated samples and the generator ids helps prevent separate generators from learning the same submanfiolds of data and experiencing the same issues of a single generator model. It is important to note that even without the MI term in the objective, the generators are still "able" to learn the disconnected support correctly. However, since the optimization is non-convex, using the MI term to explicitly encourage disjoint supports for separate generators can help avoid undesirable local minima. We show the importance of MI term in practice by removing the term from the generator objective of DMWGAN-PL, we call this variant DMWGAN-PL-MI0.

|  (a)  |  (b)  |  (c)  |  (d)  |

Figure 11: (a, b) shows DMWGAN-PL-MI0 (without MI), at 30K and 500K training iterations respectively. (c, d) shows the same for DMWGAN-PL (with MI). See how MI encourages generators to learn disjoint supports, leading to learning the correct disconnected manifold.

|  (a) Intra-class variation  |  (b) Sample quality  |

Figure 12: MNIST experiment showing the effect of mutual information term in DMWGAN-PL. (a) Shows intra-class variation. Bars show the mean square distance (MSD) within each class of the dataset for real data, DMGAN-PL model, and DMWGAN-PL-MI0 model, using the pretrained classifier to determine classes. (b) Shows the ratio of samples classified with high confidence by the pretrained classifier, a measure of sample quality. We run each model 5 times with random initialization, and report average values with one standard deviation intervals in both figures. 10K samples are used for metric evaluations.

Table 9: Inter-class variation measured by Jensen Shannon Divergence (JSD) with true class distribution for MNIST and Face-Bedroom dataset, and FID score for Face-Bedroom (smaller is better). We run each model 5 times with random initialization, and report average values with one standard deviation interval

| Model | JSD MNIST $\times 10^{-2}$ | JSD Face-Bed $\times 10^{-4}$ | FID Face-Bed |
|---|---|---|---|
| DMWGAN-PL-MI0 | 0.13 std 0.01 | 0.26 std 0.11 | 7.80 std 0.14 |
| DMWGAN-PL | 0.06 std 0.02 | 0.10 std 0.05 | 7.67 std 0.16 |