[Reviews · NeurIPS 2018]

Reviewer 1



After the rebuttal and discussions with the other reviewers I increased my overall rating from 4 to 5. I agree with the authors and the other reviewers that the observation that a disconnected data manifold can negatively affect GAN performance is important. However, I think the authors should have conducted more experiments to investigate if this problem really arises in practice (e.g. using lower dimensional embeddings in some feature space). I also disagree with the authors that the MSD measure in Euclidean space is a good measure, as Euclidean distance is known to be rather meaningless for images. Moreover, the experimental validation should *always* include the simple baseline where the generators are all trained jointly without the InfoGAN-loss. In the rebuttal, the authors show that this baseline performs (slightly) worse than the proposed method for the toy example, but I think it's vital to include this baseline in all experiments (especially in the celebA + LSUN experiment). I also think it is important to disentangle the effect of the increased model complexity of the proposed method from the effect of the regularization. ================================================================== # Summary This paper discusses the potential limitation of current state-of-the-art generative models in modeling probability distributions whose support is concentrated on a union of disjoint submanifolds. To circumvent this problem, the authors propose to use a simple variant of InfoGAN [1] with discrete variables. # Significance The observation that real-world data distribution might be concentrated on a union of disjoint submanifolds is important and hints at a possible limitation of current state-of-the-art models. Unfortunately, the authors do not directly verify this hypothesis but instead use it to motivate their method which appears to be identical to InfoGAN [1] with discrete latent variables (+ a learnable prior). It is also not clear why the authors need the InfoGAN architecture to get a good generative model and why the same effect could not be achieved by simply including discrete latent variables into the model (the explanation in ll. 113-114 is not really satisfying as it is not verified experimentally). # Clarity The paper is well-written and easy to follow. Including an example in ll. 17-18 is a nice start for the paper. However, this particular example is not really a good one, as it is certainly possible to transform a tree into a bird, e.g. by simply "zooming" into the tree where a bird sits. # Experiments The experiments are interesting, but the evaluation is a bit unclear. Was the MSD computed in Euclidean space or Feature space (ll. 176-185)? Euclidean space is probably not a good choice for an evaluation metric here, as the Euclidean distance between images is often meaningless. The discussion in ll. 192-202 why overfitting cannot occur is also not convincing, as it is not experimentally verified. A good evaluation metric should certainly be able to detect overfitting. The hypothesis that real world data distributions often contain multiple disconnected components should also be verified experimentally, e.g. by using lower dimensional embeddings of the distribution. The experiment on the union of celebA and LSUN bedrooms is interesting, but also a bit artificial, as here it is very clear that this particular probability distribution has multiple disjoint components. How do similar experiments on celebA and/or LSUN alone look like? # Other comments / questions * There is an extra "." in l. 11 * "no definite reason" in l. 35 should probably be replaced with "no satisfying explanation" * The discussion in ll. 99-104 is interesting, but when the generator cannot model the support of the true data distribution the realizability assumption of [2,3] also breaks down, which would probably be an even more compelling argument * The explanation that we need to "encourage these generators to each focus on a different submanifold of the real data" (ll. 113-114) is not really satisfying, as this could simply be learned by the generators. There are also no experiments in this paper that verify this claim. * The relationship between "partially covered real submanifolds" and "mode collapse" (l. 114) is completely unclear. * "corss entropy" should read "cross entropy" (l. 160) [1] Chen, Xi, et al. "Infogan: Interpretable representation learning by information maximizing generative adversarial nets." Advances in neural information processing systems. 2016. [2] Nagarajan, Vaishnavh, and J. Zico Kolter. "Gradient descent GAN optimization is locally stable." Advances in Neural Information Processing Systems. 2017. [3] Mescheder, Lars, Andreas Geiger, and Sebastian Nowozin. "Which Training Methods for GANs do actually Converge?." International Conference on Machine Learning. 2018.

Reviewer 2



The paper presents the dillema that current implicit models are continuous push-forwards of single mode distributions with connected support, thus by the mean value theorem the support of the generated data distribution is connected as well. While the universal approximation theorem allows these kinds of distribution to be approximated arbitrarily well in a weak (e.g. Wasserstein) sense, this is unlikely given the capacity of the models, and the training procedure (namely their inductive bias). The authors study this problem in a few toy and large scale experiments, and propose an alternative where the generated distribution is a particular learned mixture of implicit models. They add a variational regularization term so that different generators learn different manifolds (approximately they increase the mutual information between the sample and the id of the generator, so knowing which generator is maximally informative of the sample). Furthermore, since having a different number of mixture components than real manifolds or a nonuniform weighting of the components in the real manifold can lead to problems well elaborated in the text, they explore a simple alternative to learn a good prior distribution over the mixture components. There are some carefully designed experiments (and even metrics) to study the problem, as opposed to just showcasing performance. I find this sufficient for NIPS. The paper is extremely well written, and easy to read. My only concern is that some smaller claims are not validated by experiments, citations or theorems, and they should be fixed or removed. They are not central to the paper nonetheless, thus my good score: - 83-84: 'One should push the generator to learn a higher frequency function, the learning of which becomes more and more unstable' Figure 2 showcases that current methods fail at learning these kinds of data distributions well, not that the training becomes unstable, If the authors want to claim that training of high frequency functions becomes unstable, they should provide an experiment or a theorem to back up that claim. - 110: "... is more robust to bad initialization": why? Increasing the number of generators increases the probability of a bad initialization in one of the components of the mixture, and there's no experiment showing robustness to a bad initialization of one or some of the mixture components. - 133: 'leaving the loss intact': the mutual information term actually changes the loss, and potential problems are reflected in 143-145 (before the prior is learned). The regularization in the MI component and the loss for the prior are thus part of the generator's loss, and then the argument in 131-133 doesn't apply. Minor things: - Two points in line 11. - 87: This is commonly referred to as 'mode dropping', not 'mode collapse'. Mode collapse is usually referred to the case where all or almost all the samples look almost exactly the same (e.g. instead of generating only cats in a cat and dog dataset, generating only the same picture of one cat or one cat and one dog). - Question: is the distance in 183-184 in pixel space? If so, why is this a meaningful metric? - 160: typo in corss -> cross

Reviewer 3



The paper hypothesize that real images lie on a union of disjoint manifolds, but a generative adversarial network (GAN) tries to model the distribution as a single manifold and thus can not optimally learn the distribution in practice. The authors propose to use multiple generators as the solution. An EM approach is also used to learn the priors of the generators. The paper is very clear and well written. It is nicely motivated and the solution seems elegant and novel to me.The experiments are also carefully carried out. I only have a few minor comments: 1. The experiments are only on toy or small datasets. I understand that the authors claim they are not trying to achieve state-of-the-art performance, but it would be great to see if the proposed approach can really solve the difficulties in generating more complex images. 2. Maybe I miss something, but even if the generator can only learn a connected manifold, can't you also use the discriminator during inference and reject those off-manifold samples?